

# Comparative cytogenetics of microsatellite distribution in two tetra fishes *Astyanax bimaculatus* (Linnaeus, 1758) and *Psalidodon scabripinnis* (Jenyns, 1842)

Rodrigo Petry Corrêa de Sousa[1], Ivanete de Oliveira Furo[2], Gláucia Caroline Silva-Oliveira[1], Rosigleyse Corrêa de Sousa-Felix[1], Carla Denise Bessa-Brito[1], Raynara Costa Mello[3], Iracilda Sampaio[1], Roberto Ferreira Artoni[4], Edivaldo Herculano Corrêa de Oliveira[5,6] and Marcelo Vallinoto[1,7]

[1] Instituto de Estudos Costeiros, Universidade Federal do Pará, Bragança, Pará, Brazil
[2] Universidade Federal Rural da Amazônia, Parauapebas, Pará, Brazil
[3] Instituto de Ciências Biológicas, Universidade Federal do Pará, Belém, Pará, Brazil
[4] Departamento de Biologia Estrutural, Molecular e Genética, Universidade Estadual de Ponta Grossa, Ponta Grossa, Paraná, Brazil
[5] Seção do Meio Ambiente, Instituto Evandro Chagas, Ananindeua, Pará, Brazil
[6] Instituto de Ciências Naturais e Exatas, Universidade Federal do Pará, Belém, Pará, Brazil
[7] Centro de Investigação em Biodiversidade e Recursos Genéticos, Universidade do Porto, Vairão, Portugal

Corresponding author
Rodrigo Petry Corrêa de Sousa,
rodrigopcsousa@gmail.com

## ABSTRACT

**Background**. The main cytogenetic studies of the Characidae family comprise the genera *Astyanax* and *Psalidodon* involving the use of repetitive DNA probes. However, for the microsatellite classes, studies are still scarce and the function of these sequences in the genome of these individuals is still not understood. Thus, we aimed to analyze and compare the distribution of microsatellite sequences in the species *Astyanax bimaculatus* and *Psalidodon scabripinnis*.

**Methods**. We collected biopsies from the fins of *A. bimaculatus* and *P. scabripinnis* to perform cell culture, followed by chromosome extraction, and mapped the distribution of 14 microsatellites by FISH in both species.

**Results and Discussion**. The diploid number observed for both species was 2n = 50, with an acrocentric B microchromosome in *A. bimaculatus* and a metacentric B chromosome in *P. scabripinnis*. Regarding FISH, 11 probes hybridized in the karyotype of *A. bimaculatus* mainly in centromeric regions, and 13 probes hybridized in *P. scabripinnis*, mainly in telomeric regions, in addition to a large accumulation of microsatellite hybridization on its B chromosome.

**Conclusion**. Comparative FISH mapping of 14 microsatellite motifs revealed different patterns of distribution both in autosomes and supernumerary chromosomes of *A. bimaculatus* and *P. scabripinnis*, suggesting independent evolutionary processes in each of these species, representing excellent data on chromosome rearrangements and cytotaxonomy.

## INTRODUCTION

The family Characidae is the most diverse neotropical fish family, being found throughout the American continent and in Africa (*Mirande, 2019*; *Sun et al., 2021*). Currently, 1,245 valid species are known, organized into 142 genera, comprising organisms that are characterized by a small adipose fin on the caudal peduncle (*Sun et al., 2021*; *Fricke, Eschmeyer & Van der Laan, 2022*).

In this family, the genera *Astyanax* (*Baird & Girard, 1854*), with 125 species, and *Psalidodon* (*Eigenmann, 1911*), with 33 valid species, have been the two most relevant groups for studies on phylogeny, systematics, and evolution (*Terán, Benitez & Mirande, 2020*; *Silva et al., 2022*; *Tonello et al., 2022*). For a long time, *Psalidodon* belonged to the genus *Astyanax*, comprising the species included in the *Astyanax scabripinnis* complex. However, *Terán, Benitez & Mirande (2020)* proposed the validation of *Psalidodon* as a monophyletic clade, and in turn, *Astyanax* remained a polyphyletic clade.

Many lines of research have focused on the use of different markers to understand the phylogenetic relationships among Characidae species, such as morphological aspects (*Terán, Benitez & Mirande, 2020*; *Rodrigues-Oliveira, Kavalco & Pasa, 2022*), genomic DNA (*Terán, Benitez & Mirande, 2020*; *Sun et al., 2021*; *Fricke, Eschmeyer & Van der Laan, 2022*; *Silva et al., 2022*; *Tonello et al., 2022*) and cytogenetics (*Rodrigues-Oliveira, Kavalco & Pasa, 2022*; *Silva et al., 2022*; *Tonello et al., 2022*; *Sousa et al., 2023*). Among them, cytogenetics is highlighted due to the great diversity of studies involving the family, providing potential genus- and species-specific markers (*Teixeira et al., 2018*; *Cunha et al., 2019*; *Tonello et al., 2022*; *Sousa et al., 2023*).

Currently, karyotypes have been described for approximately 11 species in the genus *Astyanax* and 10 in *Psalidodon*. Nevertheless, numerous studies have been conducted to evaluate the genomic composition and cytogenetic characteristics among species in these genera (*Gavazzoni et al., 2018*; *Cunha et al., 2019*; *Schemczssen-Graeff et al., 2020*; *Silva et al., 2022*; *Tonello et al., 2022*). The substantial interest in cytogenetic research for these groups stems from the remarkable cytogenetic diversity exhibited by both genera, including multiple cytotypes, the widespread occurrence of B chromosomes in various species, natural polyploidy, and the diversity of chromosome formulas observed in these organisms (*Kavalco et al., 2009*; *Machado et al., 2012*; *Silva et al., 2022*; *Sousa et al., 2023*).

This extensive cytogenetic diversity observed in Characidae has been better understood through the use of repetitive sequence mapping, which have provided valuable information about the evolution and karyotypic diversity of this family (*Barbosa et al., 2015*; *Teixeira et al., 2018*; *Piscor et al., 2020*). However, the use of these probes in both *Astyanax* and *Psalidodon* is limited to multigene families, satellite DNAs, and histones (*Santos et al., 2013*; *Gavazzoni et al., 2018*; *Goes et al., 2022*; *Silva et al., 2022*).

Regarding the use of microsatellites, it is noteworthy that, for both genera, research is quite limited. Due to the widespread distribution of these sequences in the fish genome, such markers can provide crucial data and valuable information about the process of karyotypic differentiation for both genera. In this sense, recent studies have shown that the information obtained with the use of microsatellite probes has assisted in taxonomy,

identification of sexual systems, understanding phylogenetic relationships, population analysis, besides being used in research on genomic damage due to environmental impacts (*Cioffi et al., 2012*; *Oliveira et al., 2015*; *Yushkova et al., 2018*; *Saenjundaeng et al., 2020*; *Sousa et al., 2022*).

Considering the important of microsatellite distribution patterns in the study of chromosome evolution, our objective was to analyze and compare the distribution of these sequences in *Astyanax bimaculatus* and *Psalidodon scabripinnis*, aiming to contribute to a better knowledge of the dynamics and distribution patterns of these sequences in these two phylogenetically related genera.

## MATERIALS AND METHODS

### Specimens and chromosomal preparations

A total of three individuals (two males and one female) of the species *A. bimaculatus* were collected using a fishing net with a 25 mm mesh in the Caeté River estuary ($0°53'46.556''$S; $46°39'48.989''$W), in the municipality of Bragança (Pará, Brazil) under license ICMBIO/SISBIO, 60197/2017. The specimens collected were anesthetized and euthanized with an overdose of benzocaine (1 g/L) for the removal of biopsies from the fins. All methodological procedures and anesthesia conducts followed were approved by the National Council for the Control of Animal Experimentation (CEUA no 9847301017/2018).

The biopsies were used to stablish fibroblast cultures according to the methods of *Sasaki, Ikeuchi & Makino (1968)*, using DMEM (Dulbecco's Modified Eagle Medium) cell medium supplemented with 10% fetal bovine serum. Cell cultures were monitored daily, and flasks with 80% confluence were subjected to chromosome extraction, adopting the methodology described by *Rábová et al. (2015)*. All material from the cell culture was deposited in the cell bank of the Instituto Evandro Chagas, under the responsibility of Prof. Dr. Edivaldo Herculano Corrêa de Oliveira. Concerning *P. scabripinnis*, two samples (one male and one female) of chromosome preparations were provided by the Laboratory of Genetics and Evolution, under the supervision of Prof. Dr. Roberto Ferreira Artoni.

### Fluorescence *in situ* hybridization

Fluorescence *in situ* hybridization (FISH) experiments were performed using 14 microsatellite probes-$(CA)_{15}$, $(GA)_{15}$, $(TA)_{15}$, $(GC)_{15}$, $(CG)_{15}$, $(CAA)_{10}$, $(CAC)_{10}$, $(CAG)_{10}$, $(CAT)_{10}$, $(CGG)_{10}$, $(GAA)_{10}$, $(GAG)_{10}$, $(TAA)_{10}$, $(TAC)_{10}$-, following the procedures adopted by *Kubat et al. (2008)*, with modifications described by *Cioffi et al. (2012)*. All probes used were obtained commercially and labeled directly with Cy3 in the 5' terminal region during synthesis (Sigma, St. Louis, MO, USA).

### Microscopic analysis and image processing

A total of 30 metaphases, per experiment were analyzed to determine the diploid number, chromosome morphology, microsatellite mapping, and to assemble the karyotypes. Metaphases with better dispersion and chromosome morphology were selected for photographic recording. Images were taken in a Zeiss Axion Imager 7.2 epifluorescence microscope and analyzed with Axiovision 4.8 software (Zeiss, Jena, Germany).
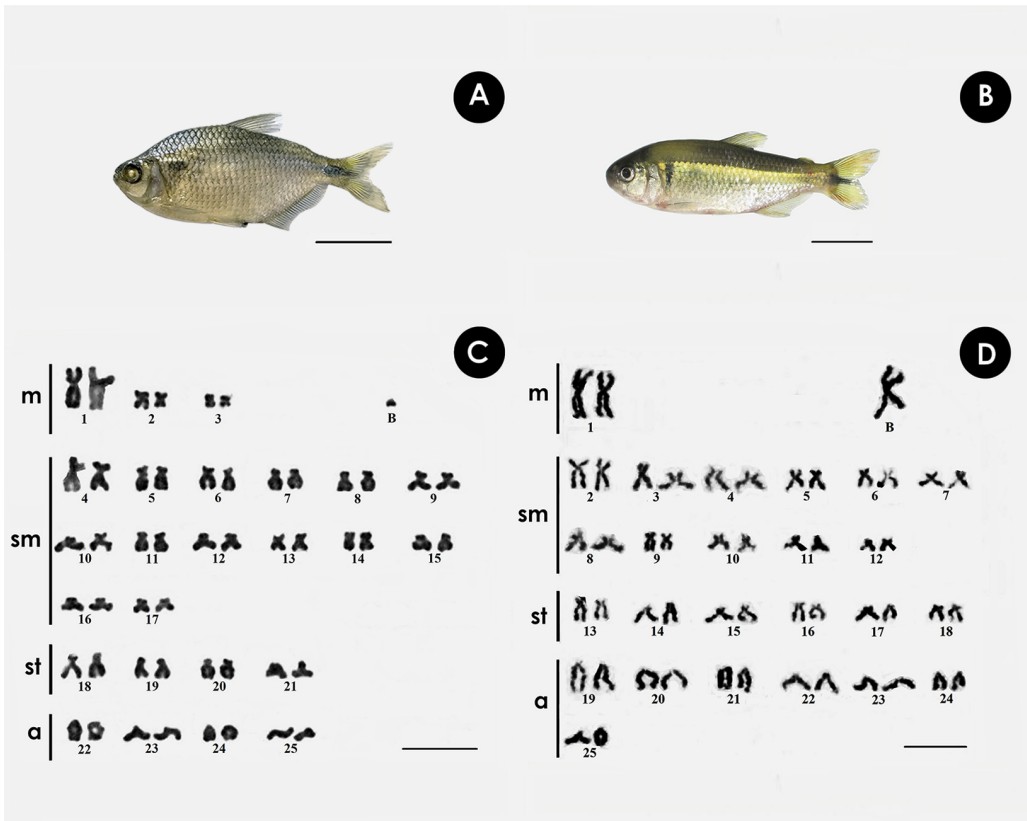

**Figure 1** **Giemsa-stained karyotype of (A, C)** *A. bimaculatus,* **highlighting the acrocentric B microchromosome; and (B, D)** *P. scabripinnis,* **highlighting the metacentric B chromosome.** Scale bar = 10 µm (C,D); 3 cm (A, B).

The karyotypes were organized using GenASIs software, version 7.2.6.19509 (Applied Spectral Imaging, Carlsbad, CA, USA). Fundamental numbers (FN) were calculated by the total number of chromosome arms, considering metacentric (m), submetacentric (sm), and subtelocentric (st) chromosomes as biarmed and acrocentric (a) as uniarmed, according to the classification proposed by *Levan, Fredga & Sandberg (1964)*.

## RESULTS

Both species have the same diploid number, with differences in chromosomal formula and FN. In *A. bimaculatus* the chromosome formula was 6m + 28sm + 8st + 8a, and FN = 92, with 1 B acrocentric microchromosome. (Figs. 1A, 1C), while the karyotype of *P. scabripinnis* was composed of 2m + 22sm + 12st + 14a, and FN = 86, with 1 B metacentric chromosome (Figs. 1B, 1D).

Chromosomal mapping of microsatellite sequences showed distinct distribution profiles for the two species. In *A. bimaculatus*, 11 microsatellite probes hybridized positively, of which $(GC)_{15}$, $(CA)_{15}$, $(CAG)_{10}$, $(CAT)_{10}$, $(GA)_{15}$, $(TAC)_{10}$, $(TAA)_{10}$, $(CAC)_{10}$, and $(GAA)_{10}$ hybridized along centromeric regions with some signals of hybridization at

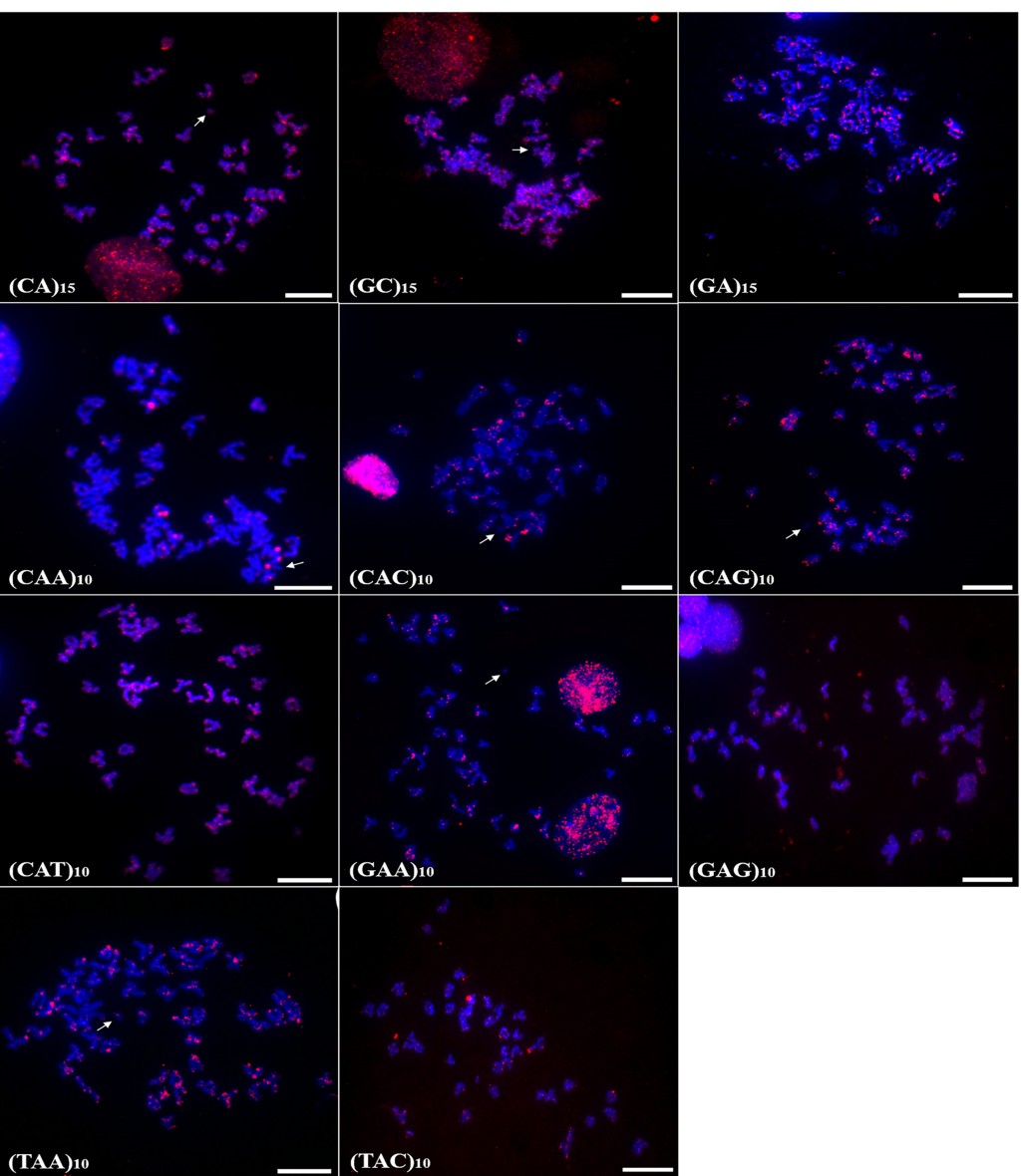

**Figure 2 Distribution of the microsatellites in the genome of *A. bimaculatus*.** The hybridization markers are in red, and the arrow indicates the chromosome B. Scale bar = 10 μm.

telomeres. Furthermore, the probes of $(GC)_{10}$, $(CAT)_{10}$, $(GAG)_{10}$, $(TAA)_{10}$, and $(GA)_{15}$ showed hybridization signals in euchromatic regions and scattered along the chromosome arms (Fig. 2).

In turn, the probe $(CAA)_{10}$ hybridized to specific regions of five chromosome pairs. Conspicuous signals of hybridization were observed on the B chromosome of *A. bimaculatus* with the $(CA)_{15}$ and $(GC)_{15}$ probes (Fig. 2).

In *P. scabripinnis* 13 microsatellite probes produced signals, with $(CG)_{15}$, $(CGG)_{10}$, $(GAA)_{10}$, $(TA)_{15}$, $(GAG)_{10}$, $(CA)_{15}$, $(CAG)_{10}$, $(CAT)_{10}$, $(GA)_{15}$, $(TAC)_{10}$, and $(CAC)_{10}$

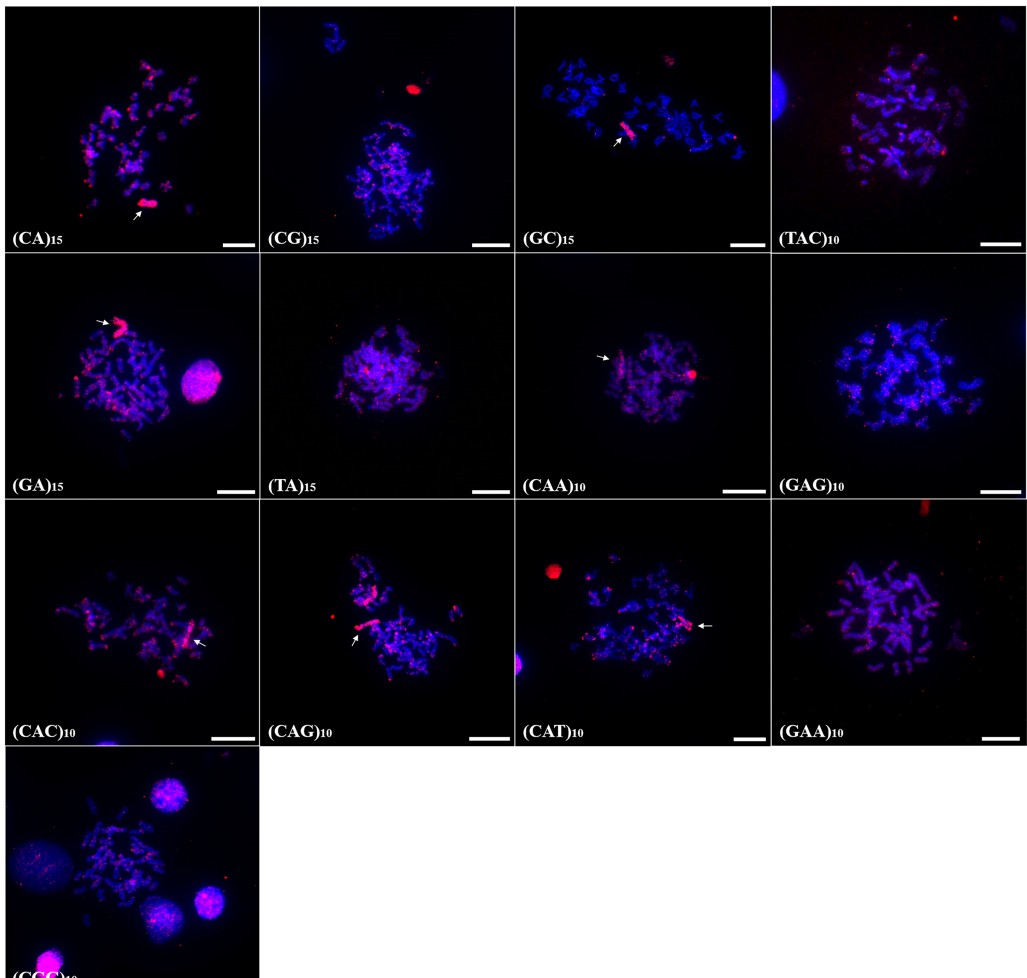

**Figure 3** **Distribution of the microsatellites in the genome of *P. scabripinnis*.** The hybridization markers are in red, and the arrow indicates the chromosome B. Scale bar = 10 μm.

hybridizing along telomeric regions, on chromosome B, and with some signals of hybridization at centromeres. In addition, probes of $(CGG)_{10}$, $(GAA)_{10}$, $(CAA)_{10}$, $(TA)_{10}$, and $(GAG)_{10}$ produced signals in euchromatic regions and scattered along the arms of the chromosomes (Fig. 3).

In turn, $(GC)_{15}$ probe hybridized on chromosome B and on the terminal portions of 5 pairs of chromosomes (Fig. 3).

## DISCUSSION

### The role of the microsatellites in the genome of *A. bimaculatus* and *P. scabripinnis*

Microsatellite DNA mapping has proven to be an excellent tool for elucidating the evolutionary dynamics of fish genomes, given the widespread presence of such repetitive sequences in eukaryotic genomes (*Bagshaw, 2017*; *Srivastava et al., 2019*). In the case of the

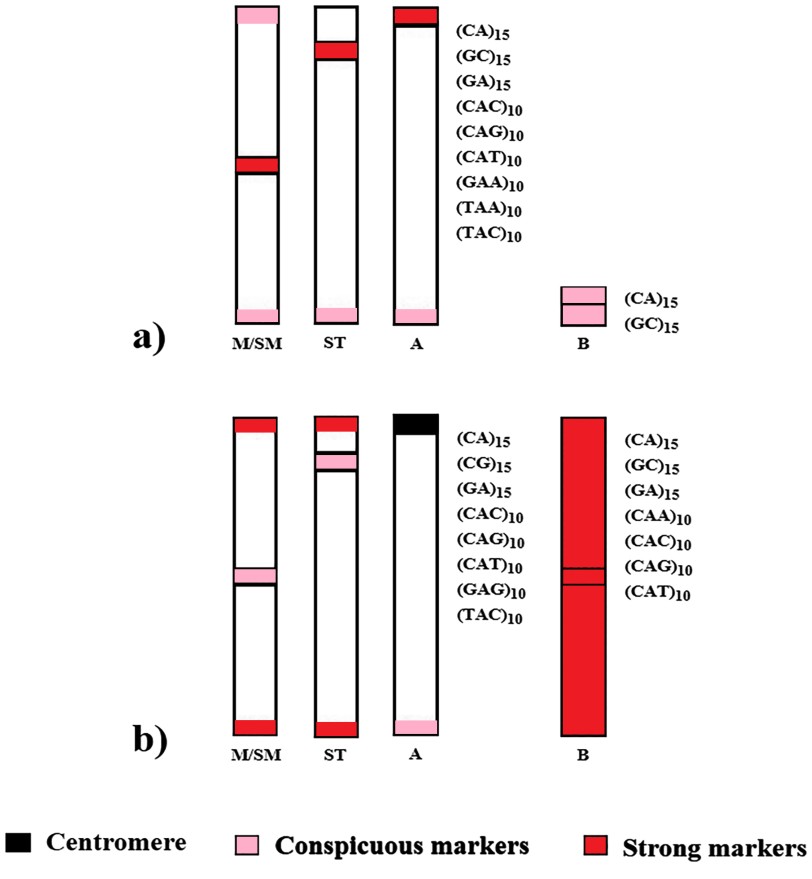

**Figure 4** Distribution scheme of the main microsatellites on autosomal and supernumerary chromosomes of (A) *A. bimaculatus* and (B) *P. scabripinnis*.

analyzed characids, the distribution patterns align with what is proposed in the literature, indicating that microsatellite sequences are more abundant in regions of low recombination rate, such as the centromeres and telomeres (*Yano et al., 2014*; *Piscor & Parise-Maltempi, 2016*; *Piscor et al., 2020*; *Sousa et al., 2022*).

Despite the phylogenetic proximity and numerous shared chromosomal features by the analyzed species, their global chromosomal hybridization of microsatellites and respective locations are distinct, suggesting independent evolution (Fig. 4). It is noteworthy that such divergences in microsatellite distribution within phylogenetically related groups have also been observed among other species of the Characidae family and in other fish groups (*Schneider et al., 2015*; *Piscor & Parise-Maltempi, 2016*; *Serrano et al., 2017*; *Sousa et al., 2022*).

These genomic differences between species indicate that the microsatellite distribution profile serves as a potential cytotaxonomic marker for the group. Furthermore, the presence of signals in euchromatic regions, observed in both species, suggests that some microsatellites may have some evolutionary purpose and could be directly associated with rearrangements (*Pathak & Ali, 2012*). In fact, chromosomal rearrangements are recurrent

findings in studies with species of the genera *Astyanax* and *Psalidodon* (*Silva et al., 2022*; *Sousa et al., 2023*), and such features may be due to the abundance of repetitive sequences present in the euchromatic regions of the chromosomes.

In general, the functions attributed to microsatellites are directly associated with structural aspects, such as chromatin organization, and DNA replication, besides developing influence in the regulation of genetic activities (*Li et al., 2002*; *Martins et al., 2005*; *Gemayel et al., 2010*). Based on the obtained results, it is suggested that a significant portion of the mapped microsatellites in both *A. bimaculatus* and *P. scabripinnis* may serve structural functions, particularly those associated with telomeres and centromeres. Additionally, some other microsatellites located in euchromatic regions, primarily trinucleotides, could potentially play a regulatory role in the genome. It is important to note that further studies employing more specific methodologies are necessary to confirm these hypotheses.

## Microsatellites distribution in the B's chromosomes of *A. bimaculatus* and *P. scabripinnis*

B chromosomes are recurrent findings in Characidae species; however, they occur most frequently in the genera *Astyanax* and *Psalidodon* (*Silva et al., 2016*; *Nascimento et al., 2020*; *Silva et al., 2022*; *Sousa et al., 2023*). In *Astyanax*, only four species have records of B's chromosomes that are always characterized by small heterochromatic acrocentric chromosomes (*Kavalco & Almeida-Toledo, 2007*; *Hashimoto et al., 2008*; *Santos et al., 2013*; *Piscor & Parise-Maltempi, 2016*; *Sousa et al., 2023*). In turn, the genus *Psalidodon* has a large number of species that have B chromosomes, which have different morphological aspects, from macro to microchromosomes (*Silva et al., 2016*; *Silva et al., 2022*).

*Silva et al. (2022)* proposed a model to explain the evolution of B chromosomes in *Psalidodon*, which can be partially applied to the genus *Astyanax*. In this model, species of the genus *Psalidodon* may have undergone different rearrangement mechanisms, leading to the different types of B chromosomes observed in the genus. However, since B chromosomes of *Astyanax* always correspond to a microchromosome, the possibility of chromosome fragmentation would be more applicable to the genus. In turn, for the analyzed *P. scabripinnis*, the hypothesis of chromatid non-disjunction, with the emergence of an isochromosome and subsequent accumulation of repetitive sequences is more plausible to justify the number of microsatellite sequences found in the B chromosome of this species (Fig. 5).

However, the reason for the limited microsatellite hybridization signals on the B chromosome of *A. bimaculatus* remains unclear. Apart from the study conducted by *Piscor & Parise-Maltempi (2016)*, which identified prominent microsatellite markings on the B chromosome of *Astyanax mexicanus*, no other species within the genus has displayed similar signals. Thus, a hypothesis can be raised to explain this trait. Although the low recombination rate in B chromosomes facilitates the accumulation of microsatellites (*Pathak & Ali, 2012*; *Silva et al., 2022*) the time for such a process in these sequences in *Astyanax* may not have been sufficient, either due to a recent emergence or a low success rate of propagation of this B chromosome in the population.

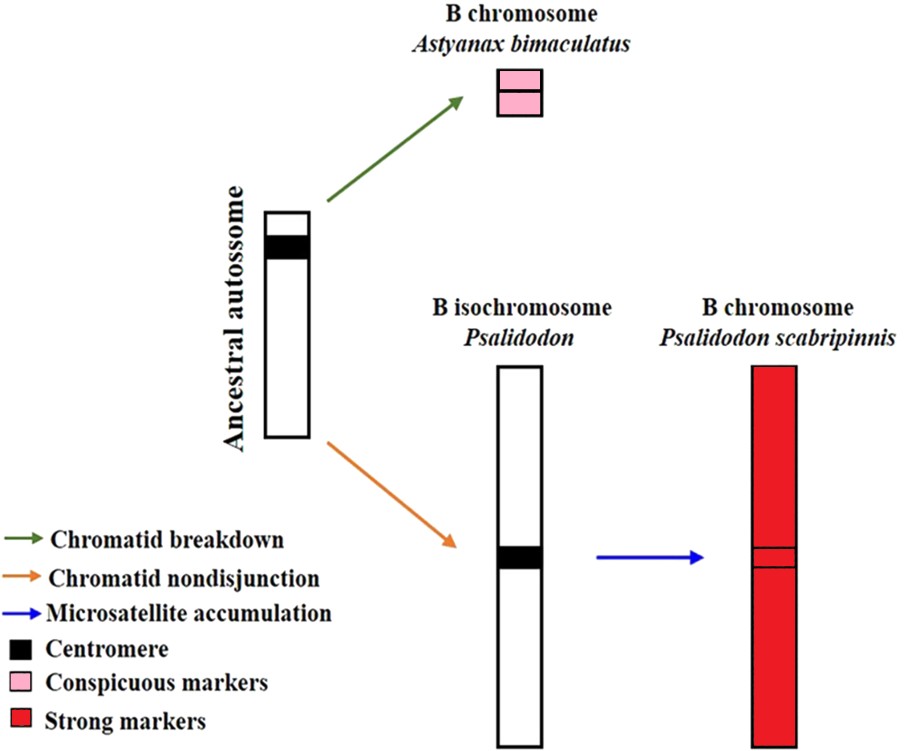

**Figure 5  Model of B-chromosome evolution of *A. bimaculatus* and *P. scabripinnis*.** Based on *Silva et al. (2022)*.

Finally, the differences in repetitive DNA content between *A. bimaculatus* and *P. scabripinnis* indicate distinct evolutionary paths for the origin of their B-chromosomes. Moreover, the variations in the distribution of microsatellites on the autosomal and supernumerary chromosomes of the two species provide valuable data on chromosomal rearrangements, as these sequences are often associated with breakpoints, which are evolutionary hotspots (*Brandström et al., 2008*; *Sousa et al., 2022*).

## CONCLUSIONS

The results of the present study contribute to the expanded understanding of the distribution and evolution of microsatellites in *A. bimaculatus* and *P. scabripinnis*, providing data that aids in comprehending karyotypic diversification at both the family and genus levels. Additionally, the comparison of microsatellite distribution allows us to infer that the composition origin of microsatellites on autosomal chromosomes and B chromosomes is different and complex for both species. These findings suggest that microsatellites may contribute to the cytogenetic diversity of *A. bimaculatus* and *P. scabripinnis*, as well as other species within the genera.

## ACKNOWLEDGEMENTS

We are grateful to the Laboratório de Evolução of the Universidade Federal do Pará and to the Laboratório de Citogenômica e Mutagênese Ambiental of the Instituto Evandro Chagas for technical support and infrastructure.

### Funding

This research was supported by the Conselho Nacional de Desenvolvimento Científico e Tecnológico (CNPq) through of the productivity grant (303889/2022-5) and the research project (407536/2021-3). Financial support was also made available to us by the Pro-Reitoria de Pesquisa e Pós-Graduação of the Universidade Federal do Pará. The funders had no role in study design, data collection and analysis, decision to publish, or preparation of the manuscript.

### Grant Disclosures

The following grant information was disclosed by the authors:
The Conselho Nacional de Desenvolvimento Científico e Tecnológico (CNPq): 303889/2022-5, 407536/2021-3.
The Pro-Reitoria de Pesquisa e Pós-Graduação of the Universidade Federal do Pará.

### Competing Interests

The authors declare there are no competing interests.

### Author Contributions

- Rodrigo Petry Corrêa de Sousa conceived and designed the experiments, performed the experiments, analyzed the data, authored or reviewed drafts of the article, technical support, and approved the final draft.
- Ivanete de Oliveira Furo performed the experiments, analyzed the data, authored or reviewed drafts of the article, technical support, and approved the final draft.
- Gláucia Caroline Silva-Oliveira conceived and designed the experiments, analyzed the data, authored or reviewed drafts of the article, technical and financial support, and approved the final draft.
- Rosigleyse Corrêa de Sousa-Felix analyzed the data, prepared figures and/or tables, authored or reviewed drafts of the article, technical and financial support, and approved the final draft.
- Carla Denise Bessa-Brito conceived and designed the experiments, performed the experiments, analyzed the data, prepared figures and/or tables, technical support, and approved the final draft.
- Raynara Costa Mello analyzed the data, authored or reviewed drafts of the article, technical support, and approved the final draft.
- Iracilda Sampaio analyzed the data, prepared figures and/or tables, authored or reviewed drafts of the article, financial support, and approved the final draft.

- Roberto Ferreira Artoni conceived and designed the experiments, analyzed the data, prepared figures and/or tables, authored or reviewed drafts of the article, technical support, and approved the final draft.
- Edivaldo Herculano Corrêa de Oliveira conceived and designed the experiments,conclusions analyzed the data, prepared figures and/or tables, authored or reviewed drafts of the article, technical and financial support, and approved the final draft.
- Marcelo Vallinoto analyzed the data, prepared figures and/or tables, authored or reviewed drafts of the article, technical and financial support, and approved the final draft.

## Animal Ethics

The following information was supplied relating to ethical approvals (*i.e.*, approving body and any reference numbers):

National Council for the Control of Animal Experimentation of the Universidade Federal do Pará.

## Field Study Permissions

The following information was supplied relating to field study approvals (*i.e.*, approving body and any reference numbers):

Universidade Federal do Pará (CEUA no 9847301017/2018).

## Data Availability

All the relevant raw data are available in the Material and Methods section and in Figs. 1, 2 and 3.

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
