# Peer review of "Comparative cytogenetics of microsatellite distribution in two tetra fishes Astyanax bimaculatus (Linnaeus, 1758) and Psalidodon scabripinnis (Jenyns, 1842)"

_PeerJ, doi:10.7717/peerj.16924_

## Round 0.1 · original submission · Minor Revisions

The manuscript is well presented and the results are well presented and discussed. However, some modifications are needed, mainly in the material and methods. See the reviewers' comments to take them into account in the revised version.

I invite you to revise and resubmit your manuscript.

Best regards,

**Language Note:** PeerJ staff have identified that the English language needs to be improved. When you prepare your next revision, please either (i) have a colleague who is proficient in English and familiar with the subject matter review your manuscript, or (ii) contact a professional editing service to review your manuscript. PeerJ can provide language editing services - you can contact us at copyediting@peerj.com for pricing (be sure to provide your manuscript number and title). – PeerJ Staff

Reviewer 1 ·

Basic reporting

Dear Editor of Peerj journal
Regarding your invitation to review the article entitled
” Comparative cytogenetics of microsatellite distribution in two tetra fishes Astyanax bimaculatus (Linnaeus, 1758) and Psalidodon scabripinnis (Jenyns, 1842)

This is a paper that without a doubt, will be of interest to scientists working in the field of molecular cytogenetic. The results of the present work expand the knowledge about the distribution and evolution of 248 microsatellites in A. bimaculatus and P. scabripinnis, generating data that assist in the 249 understanding of karyotypic diversification, both at the family and genus levels.

- Regarding this article, I consider that it has many positive points:
1. The aim of the paper is clearly defined and concisely presented.
2. Regarding the methodology used, I appreciated the multitude of tests used.
3. In general, the results are presented in a clear manner, with reference to literature data, respectively significant images.
4. Chapter of discussion is drawn up in a clear and logical manner, making reference to the relatively recent literature data.
5. The paper appears to be well written.

- However, in order to make this paper suitable for publication, minor concerns should be addressed as follow:
1. Authors must be mentioning the sex of used samples of two tetra fishes.
2. Authors must be mentioning the number of used sample in Psalidodon scabripinnis.
3. Authors should have used more from microsatellite probes.
4. In figure 1 B chromosome (B) must be mention in ligand of figure.

Final note - I recommend that this paper should be accepted for publication by introducing changes suggested to the authors. Consequently, my decision for this manuscript is Minor revision.



Yours faithfully,

Experimental design

Regarding the methodology used, I appreciated the multitude of tests used.

Validity of the findings

In general, the results are presented in a clear manner, with reference to literature data, respectively significant images.

·

Basic reporting

Clear professional English language used throughout.
Intro & background to show context. Literature well referenced & relevant.
Structure conforms to PeerJ standards, discipline norm.
Figures are relevant, high quality, well labelled & described.
Raw data supplied.

Experimental design

Original primary research within Scope of the journal.
Research question well defined, relevant & meaningful. It is stated how the research fills an identified knowledge gap.
Rigorous investigation performed to a high technical & ethical standard.
Methods described with sufficient detail & information to replica.

Validity of the findings

Meaningful replication encouraged where rationale & benefit to literature is clearly stated.
All underlying data have been provided; they are robust, statistically sound, & controlled.

Conclusions are well stated, linked to original research question & limited to supporting results.

Additional comments

MATERIALS AND METHODS
Specimens and chromosomal preparations
Please mention the number of Psalidodon scabripinnis samples. How does the author deal with it?

---

## Round 0.2 · accepted · Accept

Thank you for addressing all the reviewers' comments and improving your manuscript accordingly.

Thank you for submitting your work to this journal.

Reviewer 1 ·

Basic reporting

In my opinion, the authors addressed all the comments and now the manuscript can be accepted for publication

Experimental design

The authors addressed all the comments and now the manuscript can be accepted for publication

Validity of the findings

Yes